# Optimising Two-Stage Vacuum Heat Treatment for a High-Strength Micro-Alloyed Steel in Railway Spring Clip Application: Impact on Microstructure and Mechanical Performance

**DOI:** 10.3390/ma16144921

**Published:** 2023-07-10

**Authors:** Yao Lu, Jun Wang, Di Pan, Jian Han, Lisong Zhu, Chenglei Diao, Jingtao Han, Zhengyi Jiang

**Affiliations:** 1School of Mechanical, Materials, Mechatronics and Biomedical Engineering, University of Wollongong, Wollongong, NSW 2522, Australia; 2Welding and Additive Manufacturing Centre, Cranfield University, Cranfield, Bedfordshire MK43 0AL, UK; 3School of Materials Science and Engineering, University of Science and Technology Beijing, Beijing 100083, China

**Keywords:** Si-Cr spring steel, heat treatment optimisation, microstructure, mechanical property, fatigue performance

## Abstract

The heat treatment process is a vital step for manufacturing high-speed railway spring fasteners. In this study, orthogonal experiments were carried out to obtain reliable optimised heat treatment parameters through a streamlined number of experiments. Results revealed that a better comprehensive mechanical performance could be obtained under the following combination of heat treatment parameters: quenching temperature of 850 °C, holding time of 35 min, medium of 12% polyalkylene glycol (PAG) aqueous solution, tempering temperature of 460 °C, and holding time of 60 min. As one of the most important testing criteria, fatigue performance would be improved with increasing strength. Additionally, a high ratio of martensite to ferrite is proven to improve the fatigue limit more significantly. After this heat treatment process, the metallographic microstructure and mechanical properties satisfy the technical requirements for the high-speed railway practical operation. These findings provide a valuable reference for the practical forming process of spring fasteners.

## 1. Introduction

In the modern high-speed railway, anti-climbing ability, especially the anti-fatigue ability of the railway elastic bar, is increasingly demanding due to the higher speeds and increased loads involved in these operations [1,2]. In spring clips, the required microstructure is typically one with a fine and uniform grain size, which provides good strength, ductility, and resistance to fatigue failure. If the grain size is too large, or if there are heterogeneities in the distribution of grain sizes, then the material has reduced strength, ductility, and fatigue life [3,4]. To ensure that the bars can withstand the repeated stress of high-speed operations without experiencing fatigue failure, a post-heat treatment process was proposed [5,6]. Reports regarding the heat treatment process for different kinds of steel have been discussed before, which include high-silicon medium carbon steel [5], quenched and partitioned commercial spring steel [7], low carbon steels [8,9,10], 51CrV4 spring steel [11], and medium carbon steel [12,13]. As a typical medium carbon steel, the most common heat treatment process for spring steel is quenching and subsequent medium temperature tempering. The determination of the pre-quenching heating temperature and holding time should take the criterion that the microstructure of the steel is fully transformed into uncoarsened austenite and the carbides are completely dissolved because the growth of austenite grains may reduce the toughness of heat-treated parts [14]. Generally, the quenching temperature is selected to be 30–50 °C higher than Ac_3_ [15].

In the quenching process, an excessive quenching speed fosters the nucleation of quenching cracks. During service, the stress concentration forms at the quenching crack when applying repeated stress, which eventually results in pre-mature fracture of parts. It has been proposed that quenching cracks significantly reduce the fatigue life of spring components [16,17,18]. Inversely, the lamellar pearlite structure takes shape due to an insufficient quenching cooling rate, within which the cementite is flake-like. The ferritic matrix may generate a significant stress concentration when stressed, which leads to the fracture of cementite in advance. Then, the nucleation and proportion of microcracks eventually lead to the failure [19]. The phase transformation temperature during the quenching and cooling process are affected by the deformation status of the materials in the austenite state, which increases the distortion energy of the primitive austenite structure, further increasing the transformation temperature, and also affecting the distribution of grain-boundary precipitates [20]. Apart from that, the quenching medium for spring steel can be selected from mechanical oil, special quenching oil, or polyalkylene glycol (PAG) aqueous solution [21]. Mechanical oil is widely utilised in industrial production, benefiting from its cost-effectiveness, but it also has some shortcomings, such as insufficient cooling capacity. Special quenching oil has the cooling ability between mechanical oil and water. It can lead to the quenched parts attaining the ideal microstructure without cracking, but its price is higher. As a kind of water-soluble quenching liquid, PAG polymer dissolves from the solution under high temperature and then adheres to the surface of the workpiece, thereby reducing the cooling speed of the workpiece. Varied cooling capacities of the quenching medium can be achieved by adjusting the concentration of the quenching liquid. This kind of quenching medium also enjoys a sound economic benefit, so it is widely applied in industrial production [22]. With an increase in tempering temperature, the carbides of spring steel undergo a series of changes: dissolution of fine strip-like carbides → nucleation of flake-like carbides → decomposition of flake-like carbides → nucleation of granular carbides → growth of carbide particles. The abovementioned change in carbides also favours the evolution of mechanical properties with a trend that the strength is weakened while the plasticity is enhanced. Furthermore, the growth of carbide particles also shorten the fatigue life of spring steels [23]. In addition, the distribution of precipitates directly affects the elastic resistance of spring steel.

This study delves into the significance of heat treatment in the production of spring clips using spring steel. It aims to provide a comprehensive understanding of the effects of heat treatment on the microstructure and mechanical properties of steel, highlighting its novelty and practical importance. In this work, the influence of factors including quenching temperature, quenching holding time, quenching medium, tempering temperature, and tempering time on the final mechanical performance was systematically investigated during the spring steel heat treatment process. The significance of this research lies in its potential to advance the understanding of heat treatment techniques for spring steel used in the railway industry. By optimising the two-stage vacuum heat treatment process, the production of spring clips is anticipated with improved mechanical properties, such as higher strength, enhanced fatigue resistance, and superior durability. This has the potential to enhance the overall performance and reliability of railway tracks, ensuring enhanced safety and operational efficiency.

## 2. Materials and Methods

### 2.1. Heat Treatment Process

Testing rods were extracted from a rotary-forging Si-Cr steel rod. The chemical composition in wt.% of the experimental steel is shown in Table 1.

As given above, to keep the spring steel with a high elastic limit and enough toughness, the general heat treatment is a two-stage process, including quenching and subsequent medium-temperature tempering. The TOB-KTL1400-IV Horizontal Tube Furnace was used to heat the specimen. The process parameters that need to be considered include quenching temperature, quenching holding time, quenching medium, tempering temperature, and tempering time (see Table 2). In this study, the quenching temperatures were selected to be 830, 850, and 870 °C. Three kinds of quenching medium (12% PAG, 14% PAG, and mechanical oil) were applied, followed by holding at the quenching temperature for 25, 35, and 45 min, respectively. Afterward, the tempering process was carried out at a tempering temperature of 420–520 °C and tempering time of 60–80 min. Due to the complex factors, the orthogonal experiments (see Table 3) were carried out to avoid a large number of tests while simultaneously obtaining an optimal scheme.

The quenching heating temperatures of the material were selected to be 30–50 °C higher than Ac_3_ with a heating rate of 10 °C/min, which were finally determined in the range 830–870 °C. This is because the ferrite cannot be completely transform into austenite at low temperatures, while higher temperatures lead to the issues of energy waste and overheating of austenite grains [24,25]. Additionally, the quenching holding time has great influence on the uniformity of sample temperature, the dissolution of carbide, and the growth of austenite grain. Considering these factors, the holding time selected in this test was in the range 25–45 min. In addition, since different quenching mediums present various cooling characteristic curves or cooling mechanisms, three types of quenching medium were selected for the investigation, which were mechanical oil, 12% PAG aqueous solution, and 14% PAG aqueous solution.

Furthermore, according to previous studies on the quenching and tempering heat treatment process, the tempering temperature is normally considered to be the most important factor that affects the final mechanical properties of spring steel. Therefore, in the design of the test scheme, more attention was given to the tempering temperature. Generally, the spring steel reaches its elastic limit when the tempering temperature is between 350 and 450 °C, and the maximum fatigue limit corresponds to the tempering temperature of 450–500 °C. Thus, in this study, the tempering temperature was initially selected between 420 and 520 °C. In the aspect of determining the tempering time, it was chosen to be 60–100 min when taking the condition of carbide precipitation, energy saving, and production efficiency into consideration. Finally, since the microstructure transformation in the test spring steel during tempering mainly occurs between 100 and 350 °C, the tempering steel was heated at a slow speed (3 °C/min) between 100 and 370 °C and at a faster speed (10 °C/min) in other temperature ranges. A diagram showing the factors considered at each stage of the heat treatment cycles is displayed in Figure 1. The orthogonal test scheme was determined based on an L_18_(6^1^, 3^6^) orthogonal table. The detailed process parameters are displayed in Table 3.

### 2.2. Material Characterisation Techniques

Material characterisation techniques were carried out in terms of mechanical tests (including tensile tests, hardness tests, impact Charpy toughness tests, and fatigue resistance tests) and microstructural observations. After each process, the rod samples were cut centerline along the rolling axis by the wire cutting machine. The observation of microstructure was concentrated on the central area of metallographic specimens by a Leica optical microscope (OM), manufactured by Leica Microsystems GmbH, a company headquartered in Wetzlar, Germany. This aims to avoid the influence of oxidation and decarbonisation that were generated in the quenching and tempering processes. These surfaces were ground by various grades of sandpapers, polished, and then etched.

The microhardness testing was carried out using a MATSUZAWA Via-S Vickers innovative automatic tester (manufactured by MATSUZAWA Co., Ltd. The company is located in Saitama, Japan). The load for detecting the experimental specimens was 500 g. The indentation time was set to be 5 s on the sample surface.

A JEOL-7001 field emission gun-scanning electron microscope (manufactured by JEOL Ltd. The company is headquartered in Akishima, Tokyo, Japan) was used to indicate the analysis of electron backscatter diffraction (EBSD) mapping, energy dispersive X-ray spectrometry (EDS), and scanning electron microscope (SEM) observation, which confers information of fracture morphology, element distribution, precipitation distribution, and some other grain information. Sample analyses were conducted by using a JEOL-7001 field emission gun-scanning electron microscope at 12 mm work distance and 6.5 nA probe current, and applied together with an Oxford Instrument Nordlys-II(S) camera and Aztec 5.0 software (manufactured by Oxford Instruments plc. The company is based in Abingdon, Oxfordshire, UK). The EBSD mapping was characterised by an area of 150 × 130 μm with a step size of 0.1 μm to largely cover more grains. All the information on crystallographic orientation was collected by applying Oxford Instrument Channel 5 (HKL) software. The Charpy impact tests were conducted by using the Instron impact test system. The impact testing specimens had dimensions of 10 mm × 10 mm × 55 mm with a 2 mm deep V-notch in the central area, according to ASTM E23 [26].

For the final performance detection of the experimental steel after different heat treatment processes, the fatigue tests were carried out in the form of cyclic tensile testing by the Instron 8801 testing machine (manufactured by Instron, a company that specializes in materials testing equipment. The company is based in Norwood, MA, USA). This equipment covers the procedures for the fatigue resistance tests under an axial force-controlled condition to acquire the fatigue strength in the fatigue regime. During the entire progress, the strains have to be selected predominately in the elastic range. To keep the uniform process parameters, the force and frequency were determined to be 13.14 kN (corresponding to 450 MPa, which was determined by the yield stress) and 20 Hz, respectively. The stress ratio was −1 due to the axial force. The testing specimen was extracted from the spring steel rod, which had dimensions of 18 × 510 mm (diameter × length). Utilising the standard ASTM E606/E606M-21 protocol [27], the cylindrical rods were fashioned into dog-bone specimens. These specimens featured a 6 mm diameter in the gauge section and a gauge length of 20 mm.

## 3. Results and Discussion

### 3.1. Microstructural Analysis of the Heat-Treated Status

Based on the orthogonal testing scheme, there are nine sets of experiments that were processed by different quenching technologies. Obviously, the quenched microstructure mainly consisted of fine acicular and lath martensite (see Figure 2). In addition, the microbands (marked by yellow arrows) along the rolling direction are exhibited in Figure 2. For medium carbon steel, the first-formed martensite during quenching was dark due to the influence of self-tempering, while later-formed martensite exhibited light colour without the influence of self-tempering. According to the JB/T 9211-2008 standard [28], quenched martensite is rated as grade 3. The fine acicular and lath martensite is normally identified by high hardness together with excellent wear resistance and tensile strength [29,30,31]. The martensite needles are rated to be grade 1 because their length is less than 15 μm. To compare the influence of quenching temperature and quenching hold time on the final microstructure, it can be observed from Figure 2a–i that the martensite structure becomes coarser and the needle length becomes longer with the increase in quenching temperature and quenching hold time. Furthermore, it is evident that the acicular martensite needles do not exhibit parallel alignment to each other. Reports [32,33] revealed that the acicular plates exhibited high misorientation angle boundaries, which were greater than 45 deg. In an austenite grain, the first-formed martensite normally runs through the entire austenite grain and splits it in half, resulting in limits to the length of martensite structures. Therefore, the later-formed martensite presents a relatively smaller grain size. The relationship between quenching temperatures/hold times and the length of martensite needles can be established through statistical analysis of the prior austenite (PA) grain size distribution. This is because the size of the lamellar martensite depends on the PA grain size: an increase in the PA grain size leads to a corresponding increase in the length of the martensite needle [34].

To further observe the microstructural evolution under different tempering conditions, the SEM graphs are shown in Figure 3. It is apparent that all the microstructures are composed of granular cementite and polygonal α ferrite, while the martensite still maintains its directional distribution and acicular shape. This is because the granular carbide precipitation formed in martensite and lath edges during the tempering process [35,36]. Except for this, the granular carbide precipitation also formed at the austenite grain boundary. Accordingly, the coherent relationship was broken, leading to the spheroidisation of sheet cementite and the growth/aggregation of granular cementite.

In addition, Figure 4 shows that granular carbides are dispersed and fine in the matrix while the size of carbides was increased with an increase in tempering temperature. To further observe the high magnification of the granular carbides, EDS detections were also applied in this study. It has been proposed that granular carbides grow with the elevation of tempering temperatures, leading to a decrease in dislocation density, and the distribution in carbide precipitation is mainly concentrated on the austenite grain boundaries [37]. This phenomenon illustrates that under a relatively lower temperature, the fine carbide precipitation is generated at austenite grain boundaries to hinder the movement of the austenite grain boundary, as to prevent the growth of austenite grains. Thus, its main function is to refine the austenite grain; the microstructure after the austenite transformation is also refined.

### 3.2. Mechanical Evolution of Specimens after Heat Treatment

The heat-treated specimens were processed into standard tensile and impact samples. The tested mechanical properties of specimens after heat treatment include ultimate tensile strength (R_m_), specified non-proportional elongation strength (R_p0_._2_), Charpy impact toughness (ak), and elongation after fracture (EL). Table 4 shows all the abovementioned mechanical properties of the heat-treated samples. In this study, intuitive analysis was applied to analyse the mechanical properties test results and calculate the average deviation and range (*R*) at each factor level. The calculated results are shown in Table 5.

The influence of each factor on tensile strength can be obtained through intuitive analysis of orthogonal test results, as shown in Figure 5. Clearly, the tempering temperature and holding time during tempering are the primary influential factors on the value of Rm. This is evident from the respective influence levels of Rm is 344 and 34 Mpa, respectively. In addition, the quenching medium also exerted a non-negligible influence on R_m_, which could reach 24 MPa. Considering the R_m_ index, a tempering temperature of no higher than 460 °C can ensure a relatively high R_m_ value, which also provides a certain safety reserve for the spring clips. In addition, under the conditions of a quenching temperature of 870 °C and a quenching medium of 12% PAG aqueous solution, R_m_ is comparatively higher than that for the other parameters. The fatigue performance of spring steel is, to some extent, correlated with its R_m_. Therefore, enhancing the Rm can have a substantial impact on prolonging the fatigue life under specific conditions [38].

Likewise, according to the intuitive analysis of Rp0.2, depicted in Figure 6, it can be observed that the tempering temperature and tempering holding time are the two factors that greatly influence the R_p_0__._2_ value in this heat treatment test, with an influence level of 301 and 34 MPa, respectively. Moreover, the quenching time and quenching medium also have a certain influence on the R_p_0__._2_ value. When the tempering temperature is 460 °C, the R_p_0__._2_ of the specimen is close to the maximum stress value for spring clips. Generally, if the ultimate service stress of the products equals one-half yield strength of the material, the S-N curve almost approaches the horizontal line. Although such an infinite life product is not economical, it is necessary to retain appropriate strength reserve for the material, so that the fatigue life of the spring clips can reach the standard requirement of 5 million times. In principle, the maximum stress of the spring fastener should be less than the yield limit of the material. However, the maximum stress always occurs on the surface of the spring fastener during operation. If a small plastic zone appears on the surface of the local area, the yield of the material in this area does not cause the fracture in the spring fastener, and the safe working conditions of the spring fasteners can still be ensured. Furthermore, benefitting from the strain hardening, both the yield limit and strength of the material in the plastic zone are improved. Although the appearance of the plastic zone leads to a certain residual deformation, both the deformation and the buckle pressure loss are quite small when the local plastic zone of the spring fastener is very small, which improves the fatigue life of the spring fastener.

Through the intuitive analysis of the orthogonal test results, the factors that have greater influences on the ak value are tempering temperature, quenching temperature, and quenching holding time, with an influence level of 12.08, 4.58, and 2.60 J/cm^2^, respectively (see Figure 7). The effects of quenching temperature and quenching holding time on the ak mainly result from the PA grain size, while the tempering temperature mainly affects the ak by changing the distribution, shape, and size of the precipitated carbides. When the tempering temperature is 500 °C, the ak reaches its peak value. Nevertheless, the ak value seems to be less susceptible to the tempering time and quenching medium.

Based on the intuitive analysis of the orthogonal test results, the most influential factors affecting the EL after fracture are tempering temperature and tempering holding time. Their influence levels regarding EL are calculated to be 1.77% and 0.70%, respectively. In addition, the quenching medium and quenching holding time also have an influence on the EL after fracture (Figure 8).

From the analysis of the experimental results, the EL value after fracture of the materials tempered by different processes all acquired a sound level, which is more than 9%. Furthermore, according to the Vicker hardness results after quenching exhibited in Table 4, the sample quenched by PAG aqueous solution possesses a significantly higher hardness value than that for the sample quenched by mechanical oil. The hardness of the quenched samples increases with the elevation of quenching temperature, while the quenching holding time has little effect on the hardness after quenching. According to the relevant data in Table 5, among the designed experimental parameters, the tempering temperature is the factor that has the greatest influence on the hardness value for the tempered samples. Among them, the hardness values of samples obtained at 440 and 460 °C are reasonable, while other factors have no obvious influence on the hardness values.

From what has been discussed above, the tempering temperature is often the most influential factor on the mechanical properties index, while other factors have varied influences on the mechanical properties, which is consistent with the expectations before the test design. When the tempering temperature is set as 440–460 °C, the tensile strength of the material is greater than the maximum stress value (1496 MPa) at the active service of the spring fastener. When the tempering temperature is 460 °C, the R_p0_._2_ of the material is close to the maximum stress value for the spring fastener. Therefore, a sound match of impact toughness and elongation after fracture could be achieved.

### 3.3. Crystallographic Features at the Tempered Status

To acquire more grain information under different tempering temperatures, the EBSD technique was applied for the in-depth analysis of the samples that were tempered at 420 °C (No. 8) and 500 °C (No. 11) while maintaining the same tempering time and quenching temperature. Generally, regardless of the austenitic condition, the distribution of orientation domains is comparatively fine and consistent. This is because all the generated phases follow the Kurdjumov–Sachs rules theoretically and each austenite grain can obtain 24 orientations [33].

Although the samples were processed at different tempering temperatures, the inverse pole figure (IPF) maps exhibit few correlations (see Figure 9a,d), and it is also not recommended to determine the difference between final martensite structures from the band contrast (BC) graphs (see Figure 9b,e). Nevertheless, grain boundary maps (Figure 9c,f), where the misorientation angles in the range 20–45° (traced by reddish lines) indicate the middle-angle grain boundaries (MAGBs) and the cyan region reflects the high-angle grain boundaries (HAGBs). It can be speculated that a more uniform austenite grain distribution could be obtained at a tempering temperature of 420 °C.

Furthermore, the austenite reconstruction was applied in this to further observe the information on high-temperature grains by a reliable technique [39]. It is notable in Figure 10a,b that the sample tempered at 420 °C exhibited an overall smaller austenite grain size than the sample tempered at 500 °C. This is because, at a relatively lower tempering temperature, the carbide precipitation was mainly concentrated on the austenite grain boundaries to prevent the growth of austenite grains resulting from hindering the movement of the austenite grain boundary. The larger austenite grain with a lower incidence of austenite grain boundaries has higher hardenability, leading to a larger volume fraction of martensite in the final product phases, and thereby larger strength [40]. Simultaneously, the same developing trend regarding the martensite packet can be observed in Figure 10c,d. It is noteworthy that the martensite packet exerts a significant influence on the mechanical properties. Specifically, at lower tempering temperatures, the relatively smaller austenite grains contain fewer martensite packets with a smaller size, which leads to an increase in the strength of the steels [41]. Furthermore, based on the predicted reconstruction results, the austenite grains are distributed more evenly at a lower tempering temperature, which is consistent with the outcomes obtained from the grain boundary (GB) maps given in Figure 9.

Kernel average misorientation (KAM) maps are normally applied to describe the orientation gradients with the specific range in individual grains [42]. In this study, the selection of boundaries was determined in the range 0–5° and the 5 × 5 filter KAM maps were appropriate here, as displayed in Figure 11. Local misorientation comparisons reveal that the geometrically necessary dislocations (GND) would be significantly affected due to strain portioning and heterogeneity in microstructure [43]. For the specimens tempered at different conditions, it is difficult to distinguish the colour gradient based on the KAM maps due to small deviations. Thus, in this study, it is acceptable to calculate the local misorientation when defining the limit of random misorientation as 2°. The local misorientation of each central point was then ruled by the eight neighbour points [43,44]:(1)$$\theta_{local\;}=\overset{8}{\underset{k=1}{\sum }}\theta_{k}\cdot I_{\left(\theta_{k}<\emptyset \right)}/\overset{8}{\underset{k=1}{\sum }}I_{\left(\theta_{k}<\emptyset \right)}$$
where *θ_n_* represents the misorientation between this central point and its neighbour point *n*, ∅ is the misorientation threshold (2°), and the indicator function is defined as $I_{\left(\theta_{k}<\emptyset \right)}$. The simple method from strain gradient theory was applied to extrapolate GNDs [45]:(2)$$\rho^{GND}=\frac{2\theta }{ub}$$
where *θ* refers to the local misorientation, *u* represents the mapping unit length (step size), and *b* is Burger’s vector (BCC: 0.248 nm).

Apparently, the value for GND density was higher on the 420 °C tempered specimens than that on the 500 °C tempered counterparts, which could be attributed to the fact that the intense carbide precipitation would result in the decrease in dislocation density at a higher tempering temperature. This kind of phenomenon also validates the EDS results mentioned above.

### 3.4. Fatigue Performance Comparison

Since the tempering temperature is normally the most influential factor on the mechanical properties index, Nos. 8, 11, and 16 (corresponding to Table 3) were selected for final fatigue experiments. After fatigue testing, the fatigue cycles for Nos. 8, 11, and 16 were 309,742; 57,335; and 5007, respectively. There was an interesting finding: although No. 16 presented relatively better mechanical properties, No. 11 still has higher fatigue limits than those for No. 16.

The typical fatigue fractures are displayed in Figure 12. The crack deflection could be easily achieved in all three specimens (see Figure 12a,c,e), while the crack branching is only obtained from Nos. 11 and 16 (Figure 12c,e). Overall, the fracture in No. 8 is smoother than those in Nos. 11 and 16. In addition, the crack propagation direction is more easily observed in No. 8 from Figure 12b. For the Nos. 11 and 16 specimens, the fracture surface is relatively more ragged. Furthermore, the fracture surface for all the specimens is characterised by fatigue striations and secondary cracks. However, for Nos. 11 and 16, more fatigue striations can be observed, and the secondary cracks are also more evident. The observation of secondary cracks is coincident with the crack branching given in Figure 12c,e.

In addition, EBSD mapping was carried out for further analysis of the microstructure adjacent to the fatigue cracks, as shown in Figure 13. From the IPF maps (Figure 13a,b), a more uniform distribution of the microstructures was achieved for Nos. 8 and 16, specimens that are consistent with the GB maps given in Figure 13j–l. As mentioned above, the specimen presents a higher strength with uniform microstructural distribution; thus, it can be assumed that the specimen with a relatively higher strength (No. 8) results in a better fatigue limit. However, No. 16 also exhibited higher strength in comparison to No. 11, while it is with poor fatigue cycles. There is an interesting finding from the band contrast maps (Figure 13d–f): the ratio of martensite (dark colour) to ferrite (light colour) is much larger in No. 8 than that in Nos. 11 and 16. It can be speculated that the ratio of martensite to ferrite influences the fatigue limit more significantly. Since all specimens were conducted in the cyclic tension process until failure, several subgrains could be achieved and accumulated at the crack surfaces (Figure 13g–i).

## 4. Conclusions

The spring fastener is one of the key parts of the rail track, which provides enough buckle pressure to maintain track gauges. A high-performance spring fastener can greatly improve railway-running safety. The main research work and achievements of this study are listed as follows:(1)The influence of main process parameters on the microstructure and mechanical properties of the investigated spring steel during quenching and subsequent tempering was systematically studied, and an optimal heat treatment process was determined (quenching temperature of 850 °C, quenching holding time of 35 min, quenching medium of 12% PAG aqueous solution, tempering temperature of 460 °C, tempering holding time of 60 min).(2)The tempering temperature has a more significant influence on the microstructures and mechanical properties of the experimental steel during the heat treatment process. The granular carbides grow with the increase in tempering temperatures, which leads to a decrease in dislocation density.(3)According to the fatigue testing results, it is obvious that the specimen with a high fatigue limit has a flat and ragged fracture surface, and the microstructure after fatigue testing is more uniform. The high ratio of martensite to ferrite improves fatigue performance significantly.(4)The research conducted in this study has a certain guiding effect on the thermal processing technology of high-speed railway spring fasteners.

## Figures and Tables

**Figure 1 materials-16-04921-f001:**
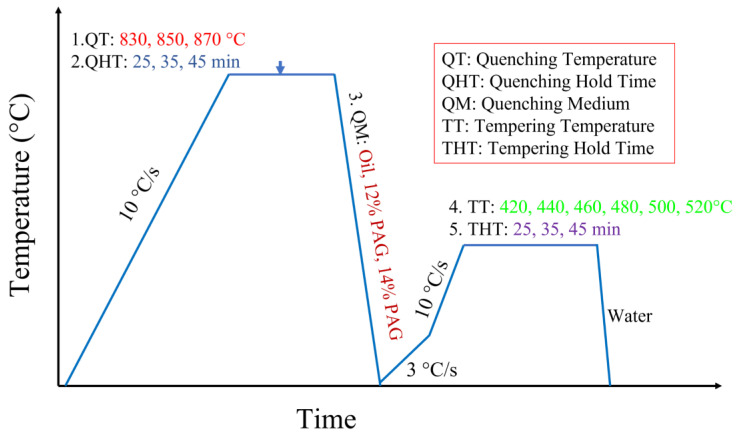
Diagram showing the factors considered at each stage of the heat treatment cycles.

**Figure 2 materials-16-04921-f002:**
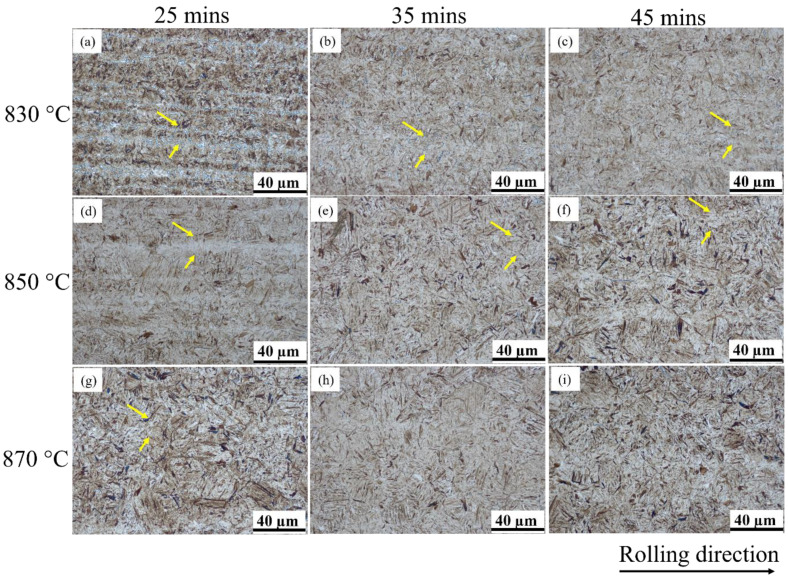
Microstructures after quenching with different quenching temperatures and quenching hold times of (**a**) 830 °C and 25 min; (**b**) 830 °C and 35 min; (**c**) 830 °C and 45 min; (**d**) 850 °C and 25 min; (**e**) 850 °C and 35 min; (**f**) 850 °C and 45 min; (**g**) 870 °C and 25 min; (**h**) 870 °C and 35 min; (**i**) 870 °C and 45 min.

**Figure 3 materials-16-04921-f003:**
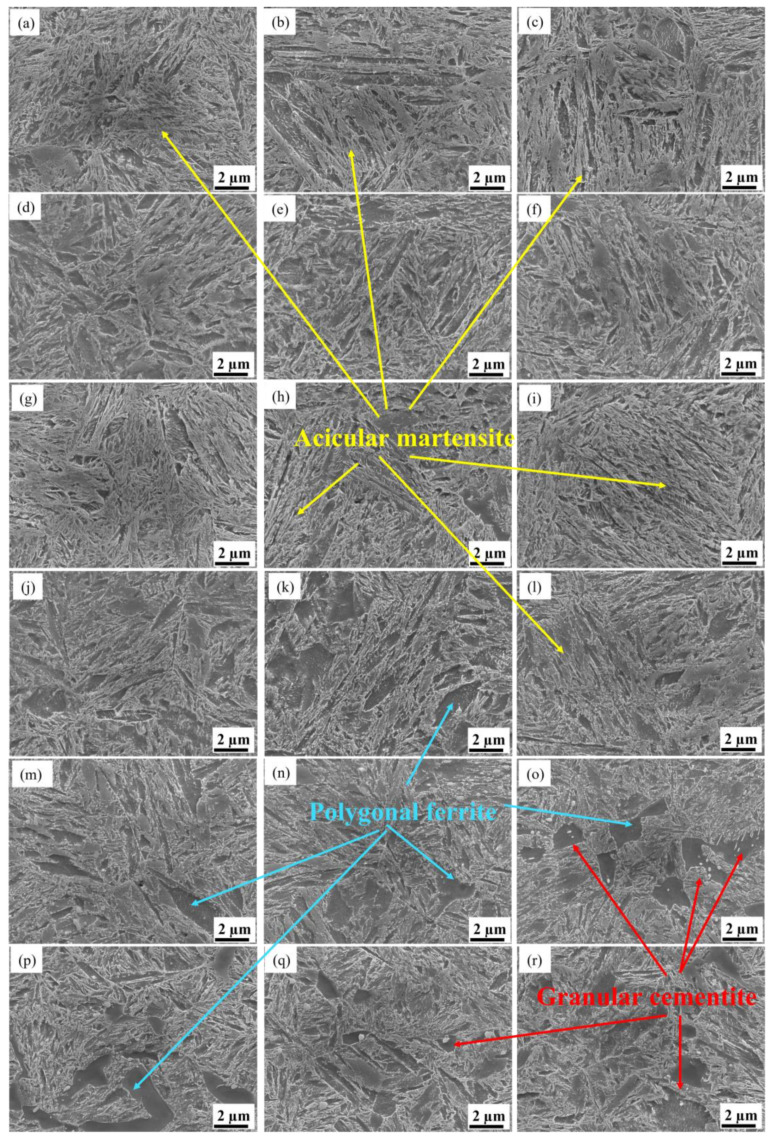
SEM observation of the tempering microstructures corresponding to the orthogonal testing scheme: (**a**–**r**) corresponds to (1)–(18).

**Figure 4 materials-16-04921-f004:**
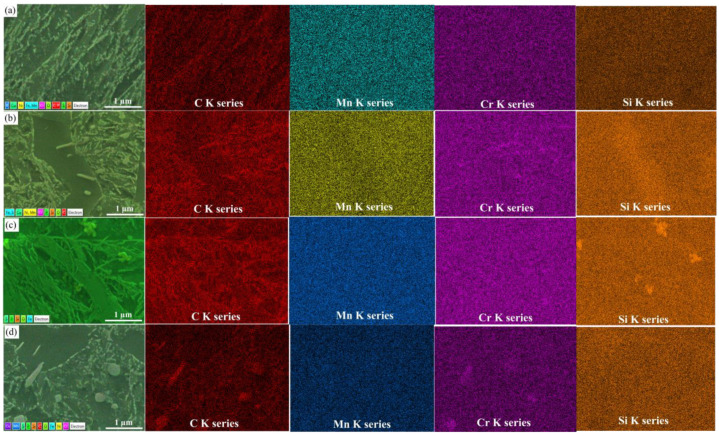
EDS mappings of the microstructures corresponding to the tempering temperature (**a**) 420 °C; (**b**) 460 °C; (**c**) 500 °C; (**d**) 520 °C.

**Figure 5 materials-16-04921-f005:**
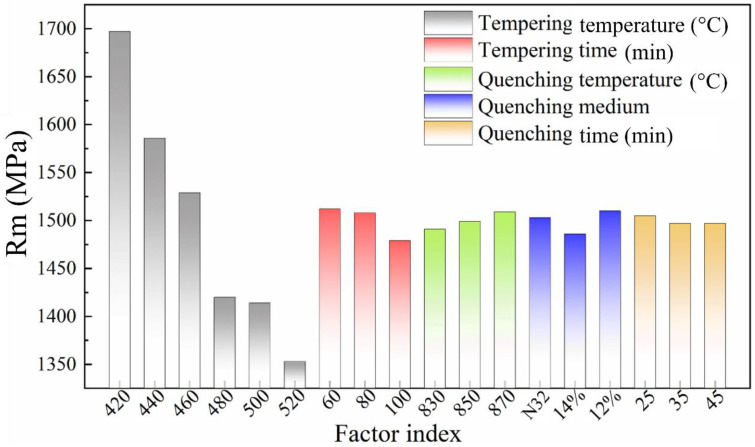
Influence of various factors on ultimate tensile strength (R_m_).

**Figure 6 materials-16-04921-f006:**
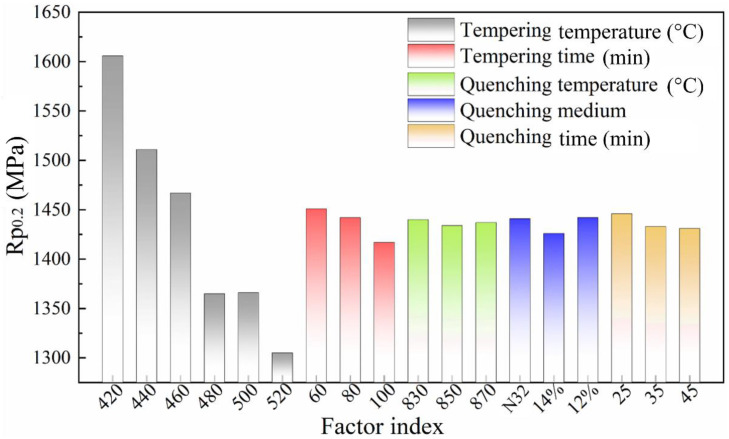
Influence of various factors on specified non-proportional elongation strength (R_p0_._2_).

**Figure 7 materials-16-04921-f007:**
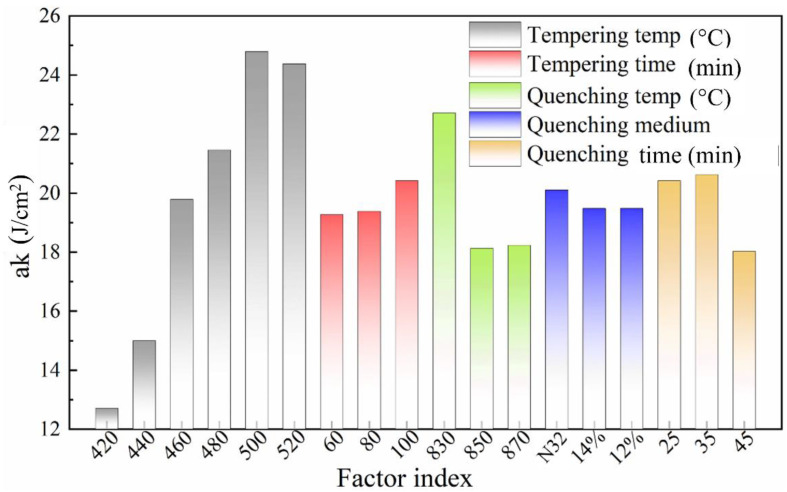
Influence of various factors on Charpy impact toughness (ak).

**Figure 8 materials-16-04921-f008:**
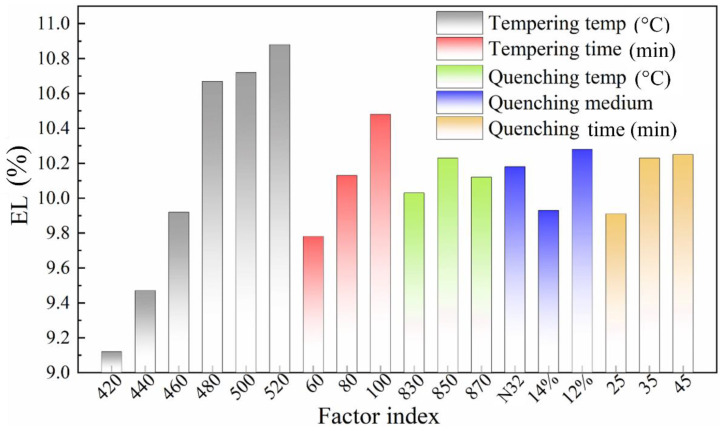
Influence of various factors on elongation (EL).

**Figure 9 materials-16-04921-f009:**
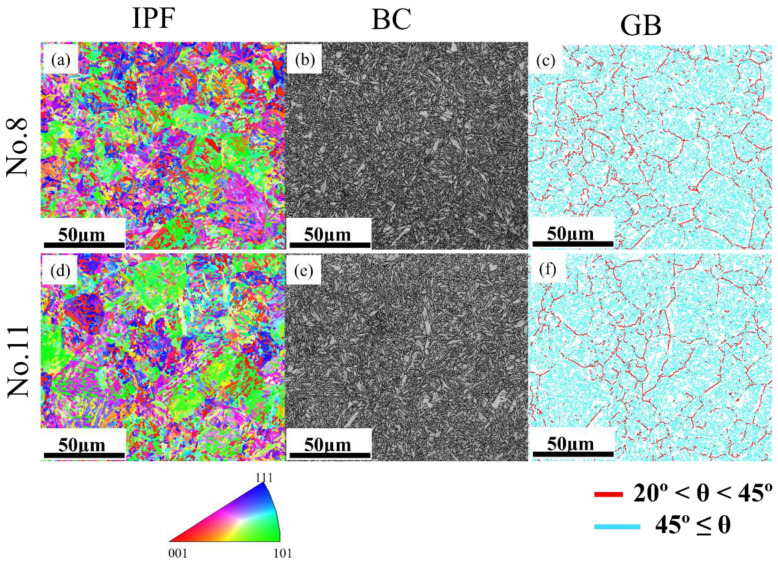
Inverse pole figures (IPF), band contrast maps, and grain boundary maps corresponding to the orthogonal testing numbers (**a**–**c**) No. 8; (**d**–**f**) No. 11.

**Figure 10 materials-16-04921-f010:**
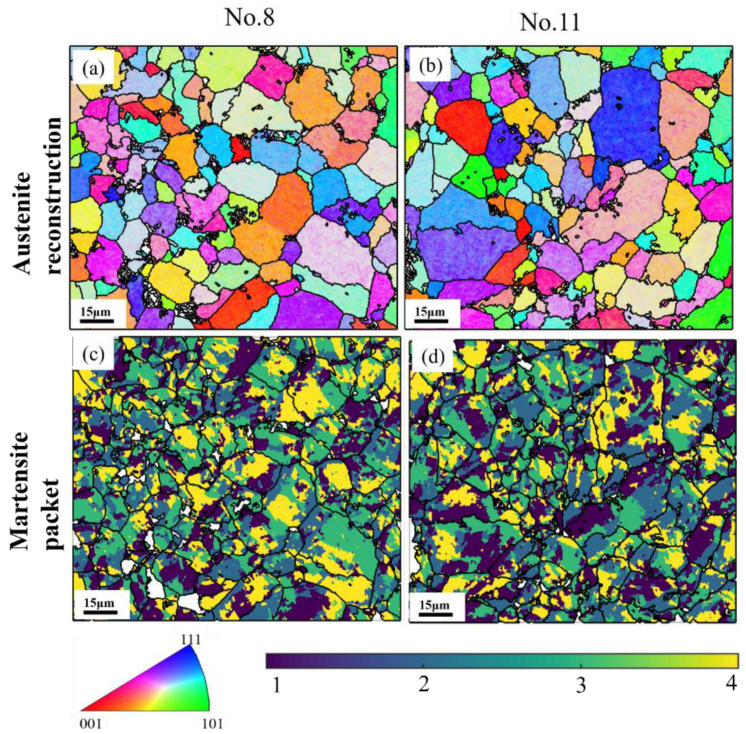
Austenite reconstruction and martensite packet distribution of the samples tempered at (**a**,**c**): 420 °C, No. 8; (**b**,**d**): 500 °C, No. 11.

**Figure 11 materials-16-04921-f011:**
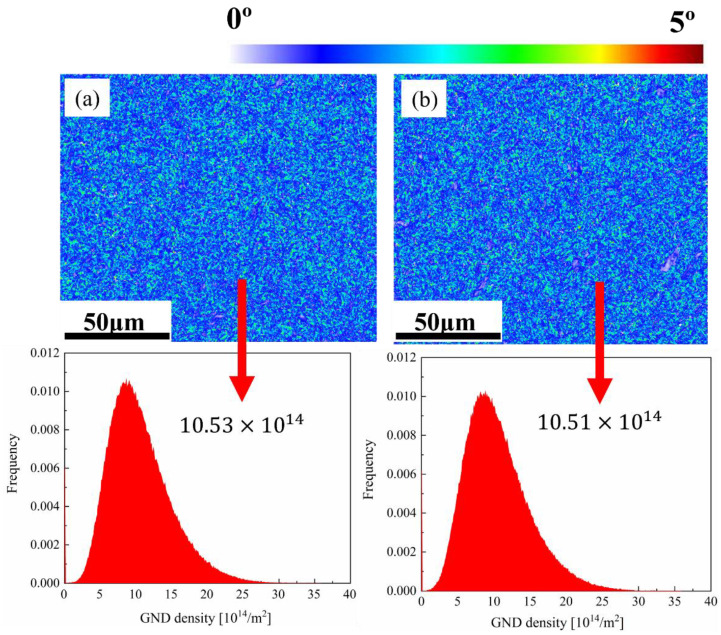
KAM maps and GND density distribution of the samples tempered at (**a**) 420 °C, No. 8; (**b**) 500 °C, No. 11.

**Figure 12 materials-16-04921-f012:**
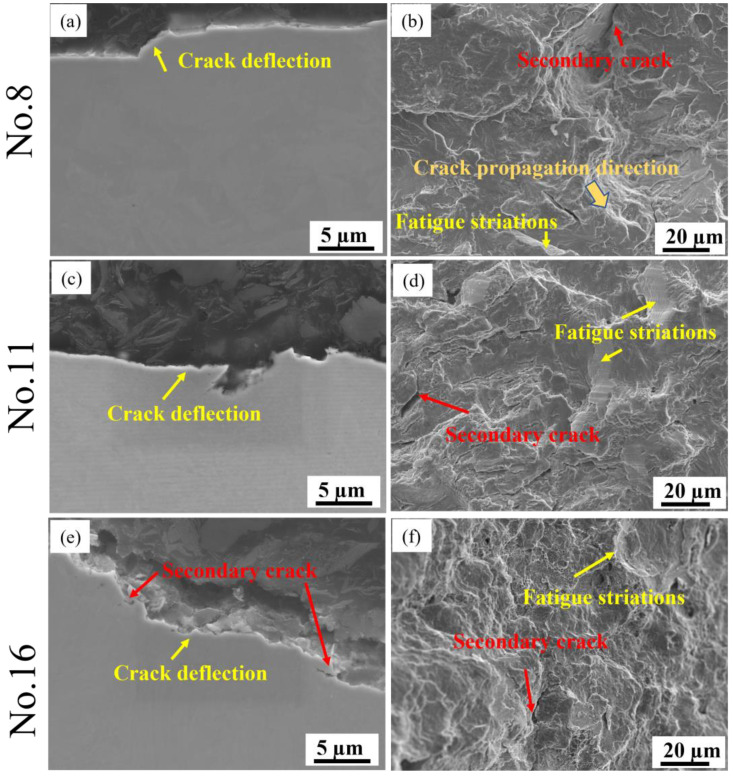
Fractographic observation of testing specimens: (**a**,**b**) No. 8; (**c**,**d**) No. 11; (**e**,**f**) No. 16.

**Figure 13 materials-16-04921-f013:**
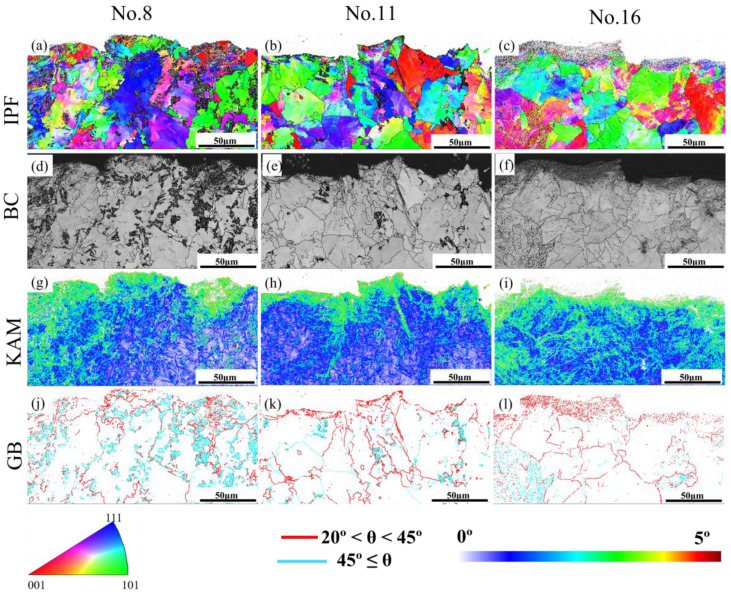
Inverse pole figures, band contrast maps, KAM maps, and grain boundary maps corresponding to specimen No. 8 (**a**,**d**,**g**,**j**); No. 11 (**b**,**e**,**h**,**k**); No. 16 (**c**,**f**,**i**,**l**).

**Table 1 materials-16-04921-t001:** The chemical composition of experimental steel (wt.%).

C	Si	Mn	Cr	Ni	Cu	S	P
0.55%	1.4%	0.65%	0.65%	0.11%	0.12%	0.012%	0.018%

**Table 2 materials-16-04921-t002:** Experimental factors for heat treatment.

Factor	Tempering Temperature (°C)	Tempering Time (min)	Quenching Temperature (°C)	Quenching Medium	Quenching Hold Time (min)
1	440	60	830	12% PAG	25
2	480	100	870	Mechanical oil	35
3	420	80	850	14% PAG	45
4	500				
5	520				
6	460				

**Table 3 materials-16-04921-t003:** Orthogonal test scheme.

No.	Tempering Tem.	Tempering Time	Quenching Tem.	Quenching Medium	Quenching Hold Time
1	1 (440 °C)	1 (60 min)	3 (850 °C)	2 (Mechanical oil)	2 (35 min)
2	1	2 (100 min)	1 (830 °C)	1 (12% PAG)	1 (25 min)
3	1	3 (80 min)	2 (870 °C)	3 (14% PAG)	3 (45 min)
4	2 (480 °C)	1	2	1	2
5	2	2	3	3	1
6	2	3	1	2	3
7	3 (420 °C)	1	1	3	1
8	3	2	2	2	3
9	3	3	3	1	2
10	4 (500 °C)	1	1	1	3
11	4	2	2	3	2
12	4	3	3	2	1
13	5 (520 °C)	1	3	3	3
14	5	2	1	2	2
15	5	3	2	1	1
16	6 (460 °C)	1	2	2	1
17	6	2	3	1	3
18	6	3	1	3	2

**Table 4 materials-16-04921-t004:** Mechanical properties under different heat treatment strategies.

Cases	R_m_ (MPa)	R_p0_._2_ (MPa)	ak (J/cm^2^)	EL (%)	Tempered Hardness (HV)	Quenched Hardness (HV)
1	1602 ± 12	1532 ± 15	13.75 ± 1.23	8.90 ± 0.65	468 ± 12	726 ± 23
2	1562 ± 8	1502 ± 10	17.50 ± 1.12	10.25 ± 0.77	468 ± 8	714 ± 14
3	1594 ± 13	1498 ± 13	13.75 ± 0.85	9.25 ± 0.58	450 ± 11	748 ± 21
4	1428 ± 9	1362 ± 9	20.00 ± 0.60	10.90 ± 0.69	445 ± 13	777 ± 15
5	1406 ± 22	1352 ± 20	22.50 ± 1.43	10.50 ± 1.02	413 ± 7	729 ± 11
6	1425 ± 18	1382 ± 12	21.88 ± 0.95	10.60 ± 1.13	440 ± 12	696 ± 9
7	1686 ± 11	1621 ± 9	16.24 ± 0.77	7.85 ± 0.34	470 ± 9	761 ± 15
8	1675 ± 12	1578 ± 11	9.38 ± 1.36	9.90 ± 0.51	500 ± 16	748 ± 13
9	1729 ± 21	1619 ± 12	12.50 ± 0.28	9.60 ± 0.37	479 ± 18	761 ± 23
10	1443 ± 16	1393 ± 13	26.88 ± 1.86	10.35 ± 1.21	417 ± 7	724 ± 20
11	1390 ± 7	1341 ± 8	23.75 ± 1.80	10.80 ± 0.59	421 ± 6	759 ± 25
12	1409 ± 14	1363 ± 23	23.75 ± 1.75	11.00 ± 0.75	426 ± 10	703 ± 15
13	1339 ± 11	1293 ± 15	18.75 ± 1.66	11.15 ± 0.91	437 ± 11	776 ± 26
14	1332 ± 19	1288 ± 12	31.88 ± 2.25	11.15 ± 1.12	426 ± 14	691 ± 6
15	1388 ± 20	1335 ± 20	22.50 ± 2.39	10.35 ± 0.54	424 ± 16	777 ± 29
16	1576 ± 19	1505 ± 17	20.00 ± 1.89	9.50 ± 0.66	461 ± 17	712 ± 16
17	1509 ± 15	1444 ± 11	17.50 ± 0.97	10.25 ± 0.48	459 ± 15	766 ± 20
18	1501 ± 8	1454 ± 7	21.88 ± 1.32	10.00 ± 0.45	475 ± 21	754 ± 8

**Table 5 materials-16-04921-t005:** Intuitive analysis of mechanical properties.

Mechanical Properties	Experiment Indexes	Factors				
		A	B	C	D	E
R_m_ (MPa)	k1j	1586	1512	1491	1510	1505
	k2j	1420	1479	1509	1503	1497
	k3j	1697	1508	1499	1486	1497
	k4j	1414				
	k5j	1353				
	k6j	1529				
	R	344	34	17	24	7
	Factors order	Primary→Secondary: A→B→D→C→E				
Rp_0.2_ (MPa)	k1j	1511	1451	1440	1442	1446
	k2j	1365	1417	1437	1441	1433
	k3j	1606	1442	1434	1426	1431
	k4j	1366				
	k5j	1305				
	k6j	1467				
	R	301	34	6	16	15
	Factors order	Primary→Secondary: A→B→D→E→C				
ak (J/cm^2^)	k1j	15.00	19.27	22.71	19.48	20.42
	k2j	21.46	20.42	18.23	20.10	20.63
	k3j	12.71	19.38	18.13	19.48	18.02
	k4j	24.79				
	k5j	24.38				
	k6j	19.79				
	R	12.08	1.15	4.58	0.63	2.60
	Factors order	Primary→Secondary: A→C→E→D→B				
EL (%)	k1j	9.47	9.78	10.03	10.28	9.91
	k2j	10.67	10.48	10.12	10.18	10.23
	k3j	9.12	10.13	10.23	9.93	10.25
	k4j	10.72				
	k5j	10.88				
	k6j	9.92				
	R	1.77	0.70	0.20	0.36	0.34
	Factors order	Primary→Secondary: A→B→D→E→C				
Tempered hardness (HV)	k1j	462	448	448	447	442
	k2j	432	447	448	452	452
	k3j	483	448	446	444	450
	k4j	422				
	k5j	429				
	k6j	465				
	R	5.0	0.2	0.2	0.8	0.7
	Factors order	Primary→Secondary: A→D→E→B, C				

## Data Availability

All data included in the current work are available upon request by contacting the corresponding author.

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
