# Peer review of "Optimising Two-Stage Vacuum Heat Treatment for a High-Strength Micro-Alloyed Steel in Railway Spring Clip Application: Impact on Microstructure and Mechanical Performance"

_materials, 2023, doi:10.3390/ma16144921_

Round 1

Reviewer 1 Report

This manuscript explored the heat treatment process that is crucial in manufacturing high-speed railway spring fasteners. By conducting orthogonal experiments, the study aimed to find the best heat treatment parameters to enhance the mechanical performance of the fasteners. The research also found that the fatigue performance, a crucial testing criterion, improved as strength increased. Additionally, a higher proportion of martensite to ferrite improved the fatigue limit considerably. After the proposed heat treatment, the resulting microstructure and mechanical properties of the spring fasteners met the technical requirements for high-speed railway use.

The study is very detailed and contains results that will contribute to the field. The following few points are recommended to be considered and authors can give details for the comments below in the manuscript.

(1) The introduction can be expanded. It is also recommended authors to mention more relevant studies about this manuscript.

How is the anti-fatigue ability of the railway elastic bar measured, and why is it becoming increasingly important in modern high-speed railways?

(2) Can you clarify what is meant by "unqualified microstructure," and how it results in low strength, plasticity, and fatigue life in spring clips? The required informations can be added in this subject.

(3) What are the effects of self-tempering on the formation and characteristics of martensite in medium carbon steel?

(4) How does the increase of quenching temperature and quenching hold time influence the coarseness and needle length of the martensite structure?

(5) Why does the first-formed martensite split the austenite grain in half? How does this impact the size of later-formed martensite?

(6) How does granular carbide growth affect the dislocation density and the distribution of carbide precipitation?

Reviewer 2 Report

Dear authors,

the research article entitled “Tailored microstructure and mechanical performance of a high-strength micro-alloyed steel through two-stage vacuum heat treatment optimization” was revised. It is focused on orthogonal experiments on heat-treated micro-alloyed steel samples. The authors affirmed that a better comprehensive mechanical performance could be obtained through this investigation, but several important revisions must be done. As a matter of fact, the obtained results must be better interconnected to each other and, at the same time, some experiments must be improved. Both microstructural and mechanical investigations must be improved. For these reasons, in this first round of revision, I recommended a major revision.

Title

Since the article aims, the title must be improved in relation to these. The authors affirmed (on page 2, lines 79-80) that investigated heat treatment processes on railway spring clip after the forming process. For this reason, the title must contain this evidence.

At the same time, more information about the railway spring clip and its forming process must be provided in the section Experimental Procedure.

Introduction

On page 1, line 56, the definition of the acronym PAG must be added. Simultaneously, acronyms “EBSD” on page 4, line 133, “SEM” on page 4, line 135, “EDS” on page 7, line 203, “IPF” on page 12, line 308, “MAGB” on page 12, line 314, “HAGB” on page 12, line 315, “GB” on page 12, line 333, “KAM” on page 13, line 337, “GND” on page 13, line 340.

A better description of the aim, novelty, and significance must be added.

Experimental procedure

1.      “Materials and Methods” is the correct name of this section in the Materials journal.

2.      The authors underlined that steel is a micro-alloyed one. In this case, chemical composition (on page 2, line 87) must be listed in a table.

3.      On page 2, line 91, please add the heating rate used during the heat treatments.

4.      On page 2, lines 93-95, please better describe that the Factors listed in Table 1 built the experiments listed in Table 2.

5.      On page 3, line 97, please better describe that 4-6 Factors are not only formed by heat treatment of tempering.

6.      On page 3, line 110, the authors refer to previous studies. Please, add references.

7.      On page 3, lines 114-122, since that heat treatments are the focal point of the present manuscript, the authors could add a temperature-time graph in which the heat treatments can be drawn.

8.      On page 4, line 126, the authors anticipate the statement “mechanical tests”. In this case, the authors must add what tests are made.

9.      On page 4, lines 127-128, fatigue samples must be discussed in this section. The authors must add the rod dimensions and the standard followed to manufacture the fatigue samples and the fatigue test.
How many samples were tested?

10.   On page 4, lines 133-134, how did the authors make to obtain element distribution and precipitation distribution through an EBSD analysis? In this case, EBSD confers information only about crystal structure and its orientation. Please, check carefully.

11.   On page 4, lines 140-143, mechanical tests are already introduced on page 4 - line 126. For these reasons, they could be joined together under a unique subchapter. I suggest 2.3. Mechanical characterisation: Charpy and Fatigue tests.
How are the impact testing specimens obtained? Did specimens were heat treated with the rod?
Please, complete the necessary information. For a better understanding, an image similar to Figure 1b can be added.

12.   On page 4, line 151, K of kilo must be a lowercase letter.

13.   On page 5, lines 155-157, letters (a) and (b) could be placed under the bottom zones of the images. The quality of Figure 1b and the position of quotes must be improved.

Results and discussion

14.   On page 5, line 163, the authors affirmed the presence of micro-bands. Please, they could be highlighted in the figures through, e.g., different arrows.

15.   On page 5, lines 167-168, add references.

16.   On page 5, the intricate sentence from lines 178 to 182 must be rewritten.

17.   Figure 2: For a better understanding of the microstructures shown in all panels, the author must add temperatures and times. Since (a)-(c) represent the quenching temperatures of 850 °C, this can be added, in the first row, to the left of panel (a). The same considerations can be done for the second and third rows for 870 °C and 850°C; and for the columns in relation to the quenching time. Please, also add the direction for rolling

18.   Figure 2 shows the microstructures of the steel after different quenching temperatures. In this case, What were the tempering temperatures? More significant conditions must be described.

19.   Figure 3: Quality must be improved. Please, report the magnification in the caption and the markers in each panel.

20.   What is the scope to consider the SEM image for each No. in Table 2? Please, only consider the more significant SEM images. In this context, a quantitative analysis in terms of cementite, ferrite and martensite can be done.
Since microstructural features are reflected in the mechanical properties, a quantitative analysis of the microstructural features must be done.

21.   On page 7, lines 192-193, please indicate these phases in Figure 3.

22.   What do the authors mean by "was improved" (on page 7, line 202)? The authors can be done a quantitative analysis as expressed in recommendation No 20.

23.   Please, improve the quality of the EDS maps shown in Figure 4. Nothing is understood in relation to element distribution.

24.   How did the mechanical properties obtain on page 7, line 217?

25.   In relation to the intuitive analysis (on page 7, lines 220-222, + Table 4), the authors must add a better description of this analysis in the section on materials and methods. Especially, the authors must also focus on the goal of this analysis. Why was it done?

26.   Table 3 lists the mechanical properties of tested samples: no associated errors are presented. Please, add these errors. In addition, how did the authors perform the Vickers microhardness/hardness tests? Please, add the information in section 2.
These values can be correlated with the microstructural features.

27.   Since HV microhardness was measured also for tempered samples, also a microstructural analysis of tempered samples must be done.

28.   On page 10, lines 245-253, the authors referred to fatigue resistance. No results were presented about stress amplitude, fatigue limit, etc. This information could be added.

29.   Crystallographic features must be replaced in the microstructural investigation.

30.   Mechanical properties are obtained in this article. Please, the author must correlate better the obtained results in terms of microstructure and mechanical properties (page 13, line 330).

31.   The authors must add evidence that is reported on page 15, lines 364-366.

32.   On page 15, lines 370-378: What are the goals if no other analysis on fatigue performance is presented?

33.   The quality of Figures 12 and 13 must be deeply improved (arrows, magnifications, ...).

Conclusions

On page 16, lines 409-411: No technical requirements are reported.

Round 2

Reviewer 2 Report

Dear authors,

thanks to follow my necessary reccomendations. At the same time, I know the relevance of SEM analsysis. In relation to the quantitative analysis, the amount of the several phases can be calculated through an adequate image analysis. As a matter of fact, the obtained mechanical properties should be correlated to the presence (%) of these several phases.